# Pesticide use safety practices and associated factors among farmers in Fogera district wetland areas, south Gondar zone, Northwest Ethiopia

**Fisseha Alebachew[1], Muluken Azage[2], Genet Gedamu Kassie[2]\*, Muluken Chanie[3]**

1 Department of Nursing, Debre Tabor Health Sciences College, Debre Tabor, Ethiopia, 2 Department of Environmental Health, College of Medicine and Health Sciences, School of Public Health, Bahir Dar University, Bahir Dar, Ethiopia, 3 Department He Informatics, Debre Tabor Health Sciences College, Debre Tabor, Ethiopia

\* geni_31280@yahoo.com

**Data Availability Statement:** All relevant data are within the paper and its Supporting Information files.

## Abstract

### Background

Farmers in developing countries, including Ethiopia, are exposed to agricultural pesticides, including pesticides that are restricted or banned in developed countries. There is little information available on pesticide use safety practices and associated factors among Ethiopian farmers, particularly in the study area.

### Objective

To assess pesticide use safety practices and associated factors among farmers in Fogera district wetland area, Amhara region, Northwest Ethiopia.

### Methods

A community-based cross-sectional study design that employs quantitative and qualitative methods was used from August 25 to September 30, 2021. Four hundred thirty participants were included by using a stratified random sampling technique. Pre-tested interview questionnaires, observational checklists, and key informant and in-depth interview guides were used to collect data. The quantitative data were collected by face-to-face interviews of farmers, whereas the qualitative data were collected by in-depth interviews of selected farmers and key informant interviews of responsible stalk holders. The data were entered into Epi data version 4.6 and analyzed using SPSS version 21. Bi-variable logistic regression was used to identify factors associated with the dependent variable. A p-value of less than 0.05 was used as a cut-off point to declare a statistically significant association between factors and outcome variables. The odds ratio and 95% CI were calculated to describe the strength of the association between factors and outcome variables. The qualitative study included 35 respondents from various backgrounds and levels of expertise, which were analyzed using thematic analysis by open-code 4.03 version software.

**Funding:** The author(s) received no specific funding for this work.

**Competing interests:** The authors have declared that no competing interests exist.

**Abbreviations:** AOR, Adjusted Odds Ratio; CI, Confidence Interval; CL, Confidence Level; DALY, Disability-Adjusted Life Years; MEDA, Moonlight Economic Development Association; NGO, Non-Governmental Organization; SPSS, Statistical Package for Social Study.

## Result

The proportion of good pesticide use safety practices in the study area was 24.4% (95% CI: 21.4%–29.3%). Educational status (adjusted odds ratio (AOR): 3.19, 95% CI: 1.44–6.71), experience of pesticide spraying (AOR: 6.85. 95% CI: 2.426–9.35), knowledge of pesticide usage (AOR: 3.40, 95% CI: 1.459–7.855), access to safety materials (AOR: 2.06, 95% CI: 1.198–3.536), and ever having received training (AOR: 4.93, 95% CI: 2.88–8.59) were factors associated with good safety practice in pesticide use. Qualitatively, limited material access, lack of government attention, insufficient training opportunities, absence of media coverage, weak enforcement of laws, and limited guideline access barred good safety practices for pesticide use.

## Conclusion

The study revealed that good safety practices were low in the study area. Being educated, having experience with pesticide spraying, having good knowledge of pesticide usage, having access to safety materials, and having received pesticide use training all increased the likelihood of good pesticide use practice. Insufficient training opportunities and material access, weak law enforcement, limited access to guidelines, and a shortage of media coverage were challenges identified qualitatively.

## Introduction

Pesticides are substance or mixture of substance used for preventing and controlling pests, weeds, vectors, rodents, and insects in agriculture to increase productivity and are also applied in the household (for mosquitoes, ticks, rats, and lice) to kill them [1]. Pesticide use safety practice justifies all procedures, actions, and policies applied to minimize the risk of exposure to potentially hazardous pesticides [2]. Pesticide use safety practices can also be demonstrated by the use of appropriate personal hygiene, effective laundry, separate pesticide storage at home, using the recommended concentration and quantity based on labeling, avoiding eating and drinking during spraying, proper use of personal protective equipment (PPE), and proper disposal of empty containers [3]. Worldwide, it is estimated that approximately 1.8 billion people engage in agriculture, and most use pesticides to increase their productivity [4]. An estimated average of 5.6 billion pounds of pesticides are utilized globally for herbicides (weeds), insecticides (insects), fungicides (fungi), and microbicides [5].

During the last two decades, international bodies have taken up the issue of pesticide utilization and adopted a number of solutions and programs to address the effects of pesticide use. Despite these efforts, global pesticide use has continued to grow steadily, reaching 4.1 million tons per year in 2017, an increase of nearly 81% from 1990 [6]. The pesticide safety practice among farmers in different countries of the world showed that 43.1% were in Nepal, 42% were in Kuwait, 50.8% were in Iran, 61% were in Uganda and Costa Rica, and 26.6% were in Ethiopia [7–11]. Following this, the global impact of inappropriate handling of pesticides led to an estimated 155,488 deaths and 7,362,493 disability-adjusted life years (DALYs) in 2016 [12].

Farmers in developing countries continue to use pesticides in increasing quantities because of ignorance of the sustainability of pesticide use, a lack of alternatives to pesticides, an underestimation of the short- and long-term effects of pesticide use, and weak enforcement of laws and regulations [6]. Pesticide importation into African countries is rapidly increasing. On the other hand, the program for controlling pesticides is limited. The reason behind this is that users have no information about the purpose of each pesticide product, the hazard level

(toxicology class), the, dosage and concentration, the method of protection, or access to protective equipment [11, 13].

The most common pesticide used in Ethiopia include organophosphates, carbamates, and to some extent organo-chlorines [14]. In Ethiopia there is no integrated poison center with a reporting system and disease-hampering estimation institutions, particularly on pesticides' effects on health and the environment of the community. This is a clear indication of a lack of concern for pesticide-related health effects and insignificant intervention in agricultural pesticide use practices [15, 16]. According to studies, farmers who had a positive attitude toward pesticide use safety practices took more precautions, used safety equipment, and used pesticides safely than those who had a negative attitude [13, 16–18]. Whatever efforts have been undertaken, pesticide users in Ethiopia in general, and the study area of Fogera wetland, in particular, are heavily exposed to short-term (e.g., skin and eye irritation, headaches, dizziness, and nausea) and long-term (e.g., cancer, asthma, and diabetes) pesticide effects [3]. Furthermore, farmers in wetland areas of Fogera district grow different products two to three times a year, and they have intensively and frequently utilized pesticides for their productivity of potatoes, onions, cabbages, and other vegetables, but there is a scarcity of information on pesticide use safety practices. Furthermore, fewer studies were conducted in Ethiopia, with a greater emphasis on pesticide use by workers on flower farms and commercial farms [19]. Therefore, the aim of the study was to assess pesticide use safety practices and their associated factors, as well as explore challenges of pesticide safety practice in the Fogera wetland area.

## Methods and materials

### Study setting and period

The study was done in Fogera Woreda Wetland from 25th August to 30th September 2021, South Gondar, Northwest Ethiopia. Fogera Woreda lies on the southeastern shore of Lake Tana on the road from Bahir Dar to Gondar, 625 km from Addis Ababa, and 55 km north of the regional capital of Bahir Dar. The district is bounded to the north by Libokemkem Woreda, to the south by Dera Woreda, to the west by Lake Tana, and to the east by Farta Woreda. The Ribb-Gumara rivers of Fogera Woreda are located on the southeastern shore of Lake Tana. Woreta and Alem Ber are two well-known towns in Fogera Woreda, with the former serving as the region's headquarters. The district has thirty-three peasant associations (PAs) or kebeles, and the city administration has five kebeles. The area is located between 11˚ 57' N and 12˚ 30' N latitude and 37˚ 35' E and 37˚ 58' E longitude. The study area/especially the wetland area has very flat land, which is known by production of rice, onion, vegetables and fruits (tomato) and Farmers have being used pesticides two to three times a year. Total annual rainfall ranges from about 1100 mm to 1530 mm/year.

### Study design and population

A community-based cross-sectional study triangulated by the qualitative study was employed by Fogera Woreda wetland farmers. All farmers who were using and applying the pesticides for agricultural practices on their agricultural farmlands at least in the last one year were included. Farmers unable to communicate due to illness during the time of data collection were excluded from the study.

### Sample size determination and sampling procedure

The required sample size for quantitative data was calculated using a single population proportion formula (n) = (Z a/2)2 P(1—P)/d2 under the following assumptions: The proportion of

pesticide use saftey practices was 21.7% (obtained from the pretested result done on Shaga Kebele in Fogera district), 95% confidence level, 5% margin of error, 10% estimated non–response rate (16), and design effect of 1.5, giving a total of 430 farmers.

Purposively, 35 participants (30 males and 5 females) were chosen for the qualitative study. Of which six, five, ten, five, and three were from Woreda and Kebele training facilitators, Kebele pesticide distribution center officials, private pesticide retailers in the town, model farmers, farmer association leaders, and NGO facilitators, respectively.

## Data collection tools and procedure

Quantitative data were collected using a pre-tested, semi-structured questionnaire derived from previous literature [10, 15, 16, 20, 21] through face-to-face interviews at home. The questionnaire was designed in English, but the interviews were conducted in the local language, (Amharic), and then converted back to English for consistency in data analysis. During the research period, five trained agricultural sector workers with a diploma were supervised by one trained BSC Environmental Health Professional. In-depth interviews and key informant interview guides were used to collect qualitative data on farmers' pesticide use safety practices. Participants were asked to focus particularly on exploring barriers to farmers' pesticide use safety practices. A standard observation checklist was also put in place to ensure farmers' safety when mixing, spraying, and disposing of empty containers.

## Data quality assurance

Data quality was assured by ensuring data collectors were trained in the data collection process. The questionnaire was first prepared in English and then, to obtain data from participants, it was translated to 'Amharic, which was the local language of the study, from which it was translated to English by experts. The Amharic version of the questionnaires was used to obtain data from participants. The data collectors were supported by supervisors and received prompt feedback. Each completed questionnaire was checked for coherence, completeness, and consistency at the same time. The daily evaluation was performed to correct any problems that could arise during the course of data collection, and the pretest was conducted on 5% of the population of Shaga Kebele, which was not selected as a study population within the study areas.

## Operational definitions

**Pesticide safety practices.** Include wearing personal protective equipment (hat, goggles, facemask, long-sleeved shirts and trousers, gloves, and boots), storing pesticides separately, and properly disposing of empty pesticide containers during pesticide handling [15].

## Data processing and analysis

Quantitative data were coded and entered into Epi-data version 4.6 statistical software. It is cleaned, edited, and analyzed using SPSS Version 21 statistical software. To explain the study population with relevant variables, descriptive statistics were used. All variables with a p-value of less than 0.25 in the bivariable logistic regression analysis were used for the multivariable analysis. P-values of less than 0.05 were considered statistically significant. Multivariable binary logistic regression analyses and adjusted odds ratios with corresponding 95% confidence levels (CL) were used to determine and report the strength of association between dependent and independent variables. For qualitative data, thematization was done, and open software version 4.03 was used. Data in the form of audio files/field notes obtained from the

participants was transcribed into the Amharic language and then translated into the English language.

## Ethical considerations

An ethical clearance was obtained from the institutional ethical review board of Bahir Dar University College of Health Science, and a supporting letter was obtained from Fogera Woreda administrative and health offices before the study started. Then informed verbal consent was obtained from the respondents after the necessary explanation about the purpose, benefits, and risks of the study by the data collectors. The data collectors continued the data collection process after they got permission from the respondents. The confidentiality of participants' information was maintained by anonymous data.

## Results

### Socio-demographic characteristics

The response rate for this study was 430 (100%). Four hundred eighteen (97.2%) respondents were males, and 344 (80%) were rural residents. Three hundred twenty-seven (76%) of the respondents were Orthodox Christians. Among all participants, 160 (37.2%) can not read and write, and 325 (75.6%) were married (**Table 1**).

### Knowledge of respondents on pesticide use safety practice

Out of 430 respondents, 315 (73.3%) had adequate knowledge about safe pesticide use practices, and those who had taken training about the safe practice of pesticide use were 171 (39.8%). Among the respondents, 238 (55.3%) of them could read pesticide labels on the containers. One hundred ninety-two (44.7%) of participants had knowledge about prohibited pesticides, and 144 (33.5%) responded that they had knowledge of guidelines for safety application. Two hundred sixty-seven (62.1%) of the involved recruits identified the route of pesticide entrance into their bodies, and 149 (34.7%) of them knew safety measures for pesticide use (**Table 2**).

### Attitude of respondents on pesticide use safety practice

Of the total 430 participants, 353 (82.1%) had a favorable attitude toward using pesticides safely. Two hundred eighty-six (53.2%) participants wanted to buy safety equipment when accessible. Respondents interested in wearing protective equipment were 352 (81.9%). Besides, 289 (67.2%) of them desired to wash their hands after spraying (**Table 3**).

### Environmental related variables

Three hundred (69.8%) of the participants took care of weather conditions while spraying, and 66 (15.3%) of them stored pesticides in a separate, dry place and closed a room, reaching out to children. Sixty-one (14.1%) respondents properly buried empty containers in the ground (Table 4).

### Safe practices of using pesticides

Out of 430 of the farmers taking part in the study, 105 (24.4%; 95% CI: 21.4%–29.3%) had good safety practices when using pesticides. Among all study subjects, 109 (25.3%) regularly used personal protective equipment, and 108 (25.2%) of them followed safety instructions while spraying pesticides. One hundred eighty-seven (43.5%) respondents said they changed

**Table 1. Socio-demographic characteristics respondents in Fogera district wetland areas, Northwest Ethiopia (n = 430).**

| Variables | Categories | Frequency (Percentage) |
|---|---|---|
| Residence | Urban | 86(20%) |
| | Rural | 344(80%) |
| Sex | Male | 418(97.2) |
| | Female | 12(2.8%) |
| Age in Years | 18–30 | 173(40.3%) |
| | 31–40 | 191(44.4%) |
| | 41–50 | 59(13.7%) |
| | >50 | 7(1.6%) |
| Marital Status | Single | 57(13.3%) |
| | Married | 325(75.6%) |
| | Divorced | 36(8.4%) |
| | Widowed | 12(2.8%) |
| Religion | Orthodox | 327(76%) |
| | Muslim | 77(17.9%) |
| | Catholic | 8(1.9%) |
| | Protestant | 9(2.1%) |
| | Other(Adventist) | 9(2.1%) |
| Educational Status | Can't read and Write | 160(37.2%) |
| | Can read and write | 78(18.1%) |
| | Primary Education | 71(16.5%) |
| | Secondary Education | 61(14.2%) |
| | Diploma and Above | 60(14%) |
| Experience with pesticide spray | < 6years | 156(36.3%) |
| | 6-10Years | 141(32.8%) |
| | >10 Years | 133(29.9%) |
| Income in months | 1500–2000 | 120(27.9%) |
| | 2001–3000 | 101(23.5%) |
| | >3000 | 209(48.6%) |
| Spraying hours worked per day | 1-4Hours | 128(29.8%) |
| | 5-8Hours | 188(43.7%) |
| | >8Hours | 114(26.5%) |
| Farm size of spray | <One hectare | 96(22.3%) |
| | One hectare | 90(20.9%) |
| | >One hectare | 244(56.8%) |
| Trend of using pesticides | Increasing | 325(75.6%) |
| | No change | 105(24.4%) |

their clothes after spraying, and 175 (40.7%) of them took a shower after spraying pesticides. Two hundred seventy (62.8%) and 119 (277.7%) participants had mixed pesticides on farm areas and used sticks while wearing gloves, respectively (**Table 5**).

## Factors associated with safety practices on pesticide use

In the bivariable logistic regression, age, educational status, having ever been exposed to pesticides before (spraying experience), income, length of time of spraying, farm size, having ever had training on pesticide use, weather conditions, the storage place of pesticides, accessibility of protective equipment, knowledge, and attitude on safety practices have a p-value of <0.25.

**Table 2. Knowledge based factors on pesticides use safety practice in Fogera district farmers of wetland area, Northwest Ethiopia.**

| Variables | Categories | Frequency(Percent) |
|---|---|---|
| Know the names of pesticides. | No | 61(14.2%) |
| | Yes | 369(85.8) |
| Know how pesticides affect human health. | No | 105(24.4) |
| | Yes | 325(75.6%) |
| Understand how pesticides affect the environment (water bodies). | No | 124(28.8%) |
| | Yes | 306(71.2%) |
| Read the pesticide labels on the container. | No | 127(44.7%) |
| | Yes | 238(55.3%) |
| Know the guidelines for the safe application of pesticides. | No | 286(66.5%) |
| | Yes | 144(33.5%) |
| Understand how pesticides enter your body. | No | 163(37.9%) |
| | Yes | 267(62.1%) |
| Know the recommended dose of pesticides on labels. | No | 268(62.3%) |
| | Yes | 162(37.7%) |
| Understand the pesticide safety precautions. | No | 281(65.3%) |
| | Yes | 149(34.7%) |
| Know to wear protective equipment while mixing and spraying. | No | 60(14%) |
| | Yes | 370(86%) |
| After spraying pesticides, change your clothes. | No | 105(24.4%) |
| | Yes | 325(75.6%) |
| Washing hands after spraying pesticides | No | 45(10.5%) |
| | Yes | 395(89.5%) |
| Take a shower after pesticide spraying. | No | 150(34.9%) |
| | Yes | 280(65.1%) |
| Know the types of prohibited pesticides. | No | 238(55.3%) |
| | Yes | 192(44.7%) |
| Take training on safe pesticide usage. | No | 259(60.2%) |
| | Yes | 171(39.8%) |
| can identify sources of information about the safety practices of pesticide use. | No | 279(64.9%) |
| | Yes | 151(35.1%) |
| Overall knowledge | Adequate | 315(73.3%) |
| | Inadequate | 115(26.7%) |

These variables were potential candidates for multiple binary logistic regressions. Educational status, spraying experience, pesticide use training, accessibility of protective equipment, and knowledge of pesticide use were associated with pesticide safety practices among these candidates as computed using multivariable binary logistic regression. Pesticide use safety practices were 3.19 times more likely among those with a diploma or higher (AOR = 3.19, 95% CI: 1.44–6.71) than among farmers who couldn't read or write. Farmers who had ever been exposed to pesticides for more than 10 years (AOR = 5.2, 95% CI: 2.43–9.35) were 5.2 times more likely to safely use pesticides than those with only 5 years of experience. When compared to farmers who had never received pesticide training, the odds of safe practices were 4.98 times higher (AOR = 4.98, 95% CI: 2.88–8.59). Farmers who had access to protective equipment (AOR = 2.06; 95% CI: 1.20–3.54) were 2.06 times more likely than those who did not have access to personal protective equipment to practice pesticide use safely. Participants who had adequate knowledge about safety practices for pesticide use (AOR = 3.40, 95% CI: 1.47–7.86)

**Table 3. Attitudes of farmers on pesticides use safety practice in Fogera district wetland areas, Northwest Ethiopia.**

| Variables | Categories | Frequency(Percentage) |
|---|---|---|
| Fear of pesticides affecting your health | Strongly disagree | 55(12.8%) |
| | Disagree | 90(20.4%) |
| | I don't know | 3(0.8%) |
| | Agree | 265(60.8%) |
| | Strongly agree | 22(5.2%) |
| Gives attention to information written on containers | Strongly disagree | 51(11.9%) |
| | Disagree | 99(23.2%) |
| | I don't know | 5(1%) |
| | Agree | 241(56%) |
| | Strongly agree | 34(17.9%) |
| Interested in wearing protective equipment | Strongly disagree | 20(4.7%) |
| | Disagree | 55(12.8%) |
| | I don't know | 3(0.7%) |
| | Agree | 310(72.1%) |
| | Strongly agree | 42(9.8%) |
| Have a positive attitude toward pesticide safety instructions. | Strongly disagree | 16(3.7%) |
| | Disagree | 75(15.2%) |
| | I don't know | 9(2.09%) |
| | Agree | 294(86.4%) |
| | Strongly agree | 46(10.7%) |
| Interested in sharing information to safely handle pesticides | Strongly disagree | 39(9.1%) |
| | Disagree | 120(27.9%) |
| | I don't know | 2(0.5%) |
| | Agree | 227(52.8%) |
| | Strongly agree | 42(9.8%) |
| Perceiving that the safe use of pesticides protects the environment | Strongly disagree | 16(3.7%) |
| | Disagree | 70(16.3%) |
| | I don't know | 4(0.9%) |
| | Agree | 241(56%) |
| | Strongly agree | 178(41.4%) |
| Interested in buying safety equipment | Strongly disagree | 77(17.9%) |
| | Disagree | 102(24.8%) |
| | I don't know | 7(1.6%) |
| | Agree | 195(45.4%) |
| | Strongly agree | 49(11.4%) |
| Interested in changing clothes after you have used them during spraying | Strongly disagree | 38(9.9%) |
| | Disagree | 99(12.3%) |
| | I don't know | 4(0.9%) |
| | Agree | 266(61.9%) |
| | Strongly agree | 23(5.3%) |

*(Continued)*

**Table 3.** (Continued)

| Variables | Categories | Frequency(Percentage) |
|---|---|---|
| Interested in washing hands after pesticide spraying | Strongly disagree | 8(1.9%) |
| | Disagree | 29(6.7%) |
| | I don't know | 2(0.5%) |
| | Agree | 326(75.8%) |
| | Strongly agree | 65(15.1%) |
| Interested in taking a shower after spraying | Strongly disagree | 20(4.7%) |
| | Disagree | 94(21.8%) |
| | I don't know | 5(1.2%) |
| | Agree | 260(60.5%) |
| | Strongly agree | 51(11.9%) |
| Overall attitude | Favorable | 353(82.1%) |
| | Unfavorable | 77(17.9%) |

were 3.40 times more likely to use pesticides safely compared with those with poor knowledge (**Table 6**).

## Qualitative finding of safety pesticide use practice

Two central themes were created that describe the safety practice of pesticide use as explored by respondents: Reasons that inhibit the use of safety equipment and methods promoting the safe practice of pesticide use Subthemes under each category include reasons inhibiting the use of safety equipment (subthemes: less attention from mass media, weak law enforcement, limited access to guidelines, insufficient availability of safety equipment, limited training opportunity, low level of understanding about the long-term effect of pesticides, the unacceptability of

**Table 4. Environmental factors on safety practices of pesticides use in Fogera district among farmers of wetland areas, Northwest Ethiopia.**

| Variables | Categories | Frequency(Percent) |
|---|---|---|
| Care of weather condition while spraying | No | 130(30.2%) |
| | Yes | 300(69.8%) |
| Place of storing pesticides | Bed room | 63(14.7%) |
| | Living room | 74(17.2%) |
| | Kitchen room | 145(33.7%) |
| | Separate room | 66(15.5%) |
| | Other | 82(19.1%) |
| Duration of storage of pesticides | 6months | 227(52.8%) |
| | 6-12months | 110(25.6%) |
| | 12-24months | 77(17.9%) |
| | Unlimited time | 16(3.7%) |
| Disposing empty containers | No | 187(43.5%) |
| | Yes | 243(56.5%) |
| If yes, how do you disposing empty containers | Burning | 27(6.3%) |
| | Burying | 61(14.2%) |
| | Leave on farm area | 147(34.2%) |

**Table 5. Practice related questions on pesticide use among farmers in Fogera district wetland areas, Northwest Ethiopia.**

| Variables | Categories | Frequency (Percentage) |
|---|---|---|
| Always use a measuring tool to add the exact amount of pesticide mentioned on the label. | No | 254(59.1%) |
| | Yes | 176(40.9%) |
| Place of mixing pesticides for spraying | Near water source | 100(23.3%) |
| | On farm areas | 270(62.8%) |
| | In the house | 60(14%) |
| Ways of mixing pesticides | With a stick but bare hands | 234(54.4%) |
| | With bare hands | 32(7.4)% |
| | With hands by wearing glove | 45(10.5%) |
| | With stick by wearing glove | 119(27.7%) |
| Type of device used for mixing pesticides | Knapsack | 362(84.2%) |
| | Bucket | 68(15.8%) |
| Regularly use protective equipment while spraying. | No | 321(74.7%) |
| | Yes | 109(25.3%) |
| Applied safety instructions on pesticide use | No | 322(74.7%) |
| | Yes | 108(25.1%) |
| Follow safety procedures while spraying. | No | 320(74.7%) |
| | Yes | 110(25.3%) |
| Check safety equipment's well-being before use. | No | 315(73.3%) |
| | Yes | 115(26.7%) |
| Change your clothes after spraying pesticides. | No | 243(56.5%) |
| | Yes | 187(43.5%) |
| After spraying, wash your hands. | No | 255(59.3%) |
| | Yes | 175(40.7%) |
| When do you take a shower after spraying pesticides | Sometimes | 122(28.4%) |
| | Always | 57(13.3%) |
| Pesticide use Safety practices score | Poor practice | 325(75.6%) |
| | Good practice | 105(24.4%) |

safety equipment, the absence of a role model, and being uncomfortable to use), and methods promoting the safe practice of pesticide use (subthemes: access to safety equipment, training opportunity, attitude change, information sharing, and encouraging model users of safety equipment).

## Theme 1: Reasons for inhibiting the safe practice of pesticide use

The problem of using safety equipment while spraying pesticides came in plenty of forms. One of the problems cited by the respondents was limited access to safety materials. A 40-year-old male farmer's association leader (participant 2) noted that: "*The Woreda agricultural office was given training on how to use safety equipment by showing the demonstration. "But they do not have access to safety materials for pesticide users."* Another farmer's association leader (participant 3) confirmed the limited access to safety equipment in such a way: "*As a solution, our farmers' association union brought safety equipment to pesticide users, but it was still not adequate."* Many farmers used their own traditional alternatives, like "**Fota" as a** hat and face mask, "**Gaunt**" as a glove, and their usual clothes of trousers and a long-sleeved shirt, as*

**Table 6. Factors associated with pesticide use safety practice showing crude odds ratio and adjusted odds ratio, Fogera district Northwest Ethiopia 2021.**

| Variables | Response categories | Safety practices of pesticide use (n = 430) | | COR(95% CI) | AOR(95% CI) |
|---|---|---|---|---|---|
| | | Poor | Good | | |
| Educational status | Can't read and write | 135 | 25 | 1 | 1 |
| | Informal education | 71 | 7 | 0.532(.22–1.291) | 0.416(0.158–1.094) |
| | Primary education(1–8) | 44 | 27 | 3.314(1.744–6.295) | 3.166(1.494–6.71)* |
| | Secondary education(9–12) | 37 | 24 | 3.0503(1.796–6.83) | 3.129(1.423–6.882)* |
| | Diploma and above | 38 | 22 | 3.126(1.589–6.15) | 3.187(1.443–7.036)* |
| Spray experience | 6month-5years | 138 | 18 | 1 | 1 |
| | 6-10years | 103 | 38 | 2.828(1.527–5.238) | 2.351(1.151–4.8)* |
| | >10years | 84 | 49 | 4.6(1.598–6.86) | 5.188(2.004–13.431)** |
| Training | No | 228 | 31 | 1 | 1 |
| | Yes | 97 | 74 | 5.611(3.465–9.085) | 4.975(2.88–8.593)** |
| Access of PPE | No | 226 | 48 | 1 | 1 |
| | Yes | 99 | 57 | 2.711(1.727–4.255) | 2.058(1.198–3.536)* |
| Knowledge | Poor | 107 | 8 | 1 | 1 |
| | Good | 218 | 97 | 5.951(2.791–12.68) | 3.397(1.469–7.855)* |

Key: * = siginificant with p-value <0.05, ** significant with p-value<0.001, 1 = reference.

*protective means.* A 28-year-old female model farmer (participant 8) described that: *"I have been using safety equipment that has been given to me by the Moonlight Economic Development Association (MEDA) training center. But most farmers tried to protect themselves by following their own experience of wind direction and a conducive time to spray ".* A 25-year-old female pesticide retailer (Participant 7) explained: *"I do not access safety equipment. "Because my clients did not ask me to bring it."*" The Woreda agricultural office and some NGOs trained us on the effects of pesticides, and we should use safety equipment when spraying pesticides," said a 35-year-old female model farmer (participant 5). *"But they do not access protective equipment at an adequate level."*" A certain number of farmers were interested in using safety measures since they had seen the effect," explained a 31-year-old male Kebele training facilitator (participant 1), "but budget constraints of the Woreda were taken as the greatest problem that handicapped access to protective equipment." Training constraints about safety measures for all pesticide sprayers are repeatedly raised by many respondents. A 28-year-old male Kebele training facilitator (participant 3) stated that: *"The Woreda agricultural office, in conjunction with some NGOs, provided training on pesticide use safety practices, but still many farmers had not received any training."* Participants also justified that ignoring law enforcement about pesticide safety practices is another restrictive factor. A 35-year-old male MEDA training facilitator commented that, *"In my view, one of the farmers' exposures to pesticide effects is weak enforcement of the law and a lack of mass media attention towards its effect."* No one forced pesticide sprayers to apply it. *"They simply spray based on their experience."* A 38-year-old male model farmer (participant 6) explained: *"In my imagination, not only poor law enforcement but also the absence of guidelines on how to apply pesticides exacerbated the level of exposure for pesticide sprayers."* A 32-year-old male Kebele pesticide distributor (participant 2) mentioned that "*no one indoctrinated pesticide sprayers in using safety materials."* "Despite the fact that there is no established system in the Fogera district for enforcing practicing safety measures*, A 30-year-old male model farmer responded that *"many farmers spraying pesticides had not accepted the use of safety equipment due to suffocation discomfort."*

### Theme 2: Methods for promoting pesticide safety practice

Law enforcement and working on behavioral change empowered safe practices. **A** 32-year-old male Woreda training facilitator (participant1) mentioned: "*I believe that pesticide use safety practices can be implemented when there is strong law enforcement and more is done on attitude change towards sprayers."* A 28-year-old male Woreda pesticide distributor (participant 2) explained that: *"Until behavioral change comes among pesticide sprayers, strong mandatory law enforcement is needed."* A 35-year-old male facilitator of the organization of rehabilitation and development in Amhara (ORDA) (participant 3) stated, "The number of farmers using safety equipment while pesticide spraying may increase when concerned government structures work with NGOs doing pesticide protection." Participants also commented that the district government offices should allocate a budget for pesticide protective material supply and access." From the time that MEDA organization gave me safety equipment, I regularly apply safety measures, and many pesticide sprayers had the greatest interest in using it if they got access," said a 25-year-old male farmer (participant 4). A 32-year-old male model farmer (participant 6) explained that: *"In the beginning, safety materials were not comfortable to use."* But now I have adopted it and do not spray pesticides without it." "By observing me, other farmers are inspired to use safety equipment as they have the chance." A 28-year-old male model farmer (participant 3) expressed that: *"Farmers in Fogera district have no problem with income to buy safety equipment." "As a result, the concerned body attempted to change farmers' attitudes and provide them with access to materials."* Respondents also emphasized the importance of training in order to advance pesticide sprayer awareness and attitude.

A 28-year-old male Kebele training facilitator (participant 2) remarked that: *"In addition to lessons learned from experience, training empowers farmers' inspiration to use safety equipment while spraying pesticides."*

A 35-year-old male model farmer (participant 6) mentioned: *"After training, I have applied complete safety equipment, including all covers."* Even I have discussed with my neighbors how beneficial it is to be free of pesticide symptoms.

## Discussion

This study revealed that the prevalence of safety practices was 24.4% (95% CI: 21.4%–29.3%). Educational status, spraying experience, ever having had training on pesticide use, accessibility of protective equipment, and knowledge of using pesticides are associated with safe pesticide use practices. The qualitative study also reported that equipment access is a crucial issue for safe practices in pesticide use. This study was consistent with the study done in southwest Showa and east Showa, Ethiopia, which found 26.6% and 28.1%, respectively [10, 22].

The finding of such a study was higher than research done in Northwest Ethiopia (8.29%) and among rice farmers in Iran (8.6%) [21, 23]. This disparity could be attributed to the time lapse between studies and the various study subjects included in the studies. However, this study had fewer participants than those conducted in Bahirdar and Gondar, Ethiopia (61.3% and 63.8%), respectively [24, 25]. This disparity could be attributed to study subject differences, organizational access to safety equipment, and having good access to training since the studies were conducted on flower farm workers.

The results of safe pesticide use practices were also lower than those of studies done in Uganda (55%), Costa Rica (61%), Iran (50.8%), Nepal (43.1%), and Kuwait (42%) [7–9, 21]. This disparity might be due to the research setting, the educational level of the study individuals, and economic and socio-demographic differences.

In this study, the educational status of a diploma and above was positively associated with safe pesticide use practices. Pesticide sprayers with diplomas and above have a safer practice

than uneducated farmers. The Southwest United States, Ethiopia, Nepal, and Nigeria [7, 9, 10] all contributed to this research. The reason for this might be that more educated farmers have prior knowledge about the toxic effects of pesticides through formal education than uneducated farmers. Furthermore, educated farmers are more likely to accept and implement changes after receiving training than uneducated farmers.

The spraying experience of farmers was also significantly associated with the safety practice of pesticide use. Farmers with more than ten years of pesticide spraying experience sprayed pesticides more safely than those with only five years of experience. It was supported by a study done in Cameroon and Iran [5, 9]. Similarly, it was supported by qualitative observation data. The justification behind this could be that farmers with longer years of pesticide spraying exposure would clearly see the effects of unsafe pesticide use. Furthermore, they would get more information about the importance of safe pesticide use from different sources during these times and could develop a greater interest in saving themselves from being vulnerable to pesticide residuals and trying to protect themselves from such bad consequences.

Pesticide training and knowledge were statistically significant for safe pesticide use practices. Farmers who received pesticide application training practiced it more safely than those who did not. It is also recognized by qualitative observational data. Such conditions were supported by a study done in Nepal [7]. The reason could be that farmers who receive pesticide safety training will raise their awareness, gain knowledge, and practice more effectively. In such a study, the accessibility of safety equipment was positively associated with safe pesticide use practices. This was supported by qualitative data. It was supported by a study done in southwest Showa, Ethiopia, and Uganda [8, 10]. Whatever pesticide sprayers had good knowledge and attitude toward safety practices and protecting themselves from pesticides, without accessibility and availability of safety materials, everything is a dream. As revealed from the qualitative study, farmers who used pesticides had acquired safety equipment from the government, NGOs, private retailers, and farmers' association distribution centers, but they were not satisfied with the accessibility of safety equipment to protect themselves from pesticide effects.

## Conclusion

The study revealed that good safety practices were low in the study area. Being educated, having been exposed to pesticides before, having good knowledge of pesticide usage, having access to safety materials, and having ever had training on pesticide use increased the odds of good practice in pesticide use. Insufficient training opportunities and material access, weak law enforcement, limited access to guidelines, and a shortage of media coverage were challenges identified qualitatively. These identified modifiable factors are the focus of interventions to strengthen and design interventions to improve pesticide use safety.

## Supporting information

**S1 Data.**
(SAV)

**S2 Data.**
(DOCX)

## Acknowledgments

The authors acknowledged Bahir Dar University, the College of Medicine and Health Sciences, and the School of Public Health for supporting and facilitating this study. The authors also

acknowledged data collectors, supervisors, and study participants for their contributions to this work.

## Author Contributions

**Conceptualization:** Fisseha Alebachew, Muluken Azage, Muluken Chanie.

**Data curation:** Fisseha Alebachew, Muluken Azage, Genet Gedamu Kassie, Muluken Chanie.

**Formal analysis:** Fisseha Alebachew, Muluken Azage, Genet Gedamu Kassie.

**Funding acquisition:** Fisseha Alebachew.

**Investigation:** Fisseha Alebachew.

**Methodology:** Fisseha Alebachew, Muluken Azage, Genet Gedamu Kassie, Muluken Chanie.

**Project administration:** Fisseha Alebachew.

**Resources:** Fisseha Alebachew.

**Software:** Muluken Azage, Genet Gedamu Kassie.

**Supervision:** Muluken Azage, Genet Gedamu Kassie, Muluken Chanie.

**Validation:** Muluken Azage, Genet Gedamu Kassie.

**Visualization:** Fisseha Alebachew.

**Writing – original draft:** Fisseha Alebachew.

**Writing – review & editing:** Muluken Azage, Genet Gedamu Kassie, Muluken Chanie.

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
