## [Decision Letter · Decision Letter 0]

12 Oct 2022

PONE-D-22-18751Pesticide use safety practices and associated factors among farmers in fogera district wetland areas, south Gondar zone, Northwest Ethiopia 2021PLOS ONE

Dear Dr. Kassie,

Thank you for submitting your manuscript to PLOS ONE. After careful consideration, we feel that it has merit but does not fully meet PLOS ONE’s publication criteria as it currently stands. Therefore, we invite you to submit a revised version of the manuscript that addresses the points raised during the review process.

We look forward to receiving your revised manuscript.

Kind regards,

Haruna Musa Moda

Academic Editor

PLOS ONE

Journal Requirements:

Additional Editor Comments:

Dear author,

Many thanks for considering PLOS One as platform to publish your work.

While the theme aligns with the journal objective, however it was concluded the work is not ready for submission hence will require further development. On this occasion a major revision has been recommended.

Best regards

Reviewers' comments:

Reviewer's Responses to Questions

**Comments to the Author**

1. Is the manuscript technically sound, and do the data support the conclusions?

Reviewer #1: Partly

Reviewer #2: Partly

2. Has the statistical analysis been performed appropriately and rigorously? 

Reviewer #1: I Don't Know

Reviewer #2: No

3. Have the authors made all data underlying the findings in their manuscript fully available?

Reviewer #1: Yes

Reviewer #2: No

4. Is the manuscript presented in an intelligible fashion and written in standard English?

Reviewer #1: No

Reviewer #2: No

5. Review Comments to the Author

Reviewer #1: A) The English Language needs to be improved upon

B) Needs to get a recent edition of the PLOS ONE Journal to guide in i) presentation of results ii) discussions

C) Adhere to the PLOS ONE Template for listing references. (Refer to https://journals.plos.org/plosone/s/submission-guidelines)

i) Appropriate URL and DOI should be properly inserted

ii) Punctuation

iii) Justification

Reviewer #2: The manuscript describes original research on Pesticide use safety practices and associated factors among farmers in fogera district wetland areas, south Gondar zone, Northwest Ethiopia. The title will need reviewing to exclude 2021. The objectives of the study were clear. Manuscript has no line numbering and therefore difficult to provide feedback on specific lines. The study is interesting and would have like to see some level of reorganisation to enhance the specific impact of the study. Generally, the manuscript contains lots of grammatical errors, and some sections are unclear and difficult to read. I strongly recommend the authors check the grammar carefully and enhance the clearness of sentences throughout the manuscript. I will advice author to, if possible, use professional English language editors, if available.

Abstract will need reviewing, particularly, on the methods/procedures of the experiment to reflect order of processes presented in the main text. Overly, statistical analysis is what is presented in the methods and not sufficient of how data was collected. Why include 2021 in the objective, which is indicated as aim in the main text. Results is unclear as presented.

In the introduction, authors have defined pesticides as drug used for controlling and preventing pests, weeds, vectors, rodents, insects. Is this definition of pesticides as drugs, right? It would also be clearer for authors to indicate the different types of pesticides used for controlling weeds, rodents etc. Most part of the introduction is unclear due to grammatical issues. At the beginning of paragraph 2 of the introduction, authors referred to ‘international bodies taking up the issues and adopting a number of solutions. It is not clear which issues authors were referring to. Can this be clarified? Authors also indicated that ‘despite the efforts, global pesticide use has continued to grow steadily to 4.1 million tons per year in 2017. Would be useful for authors to indicate from which year to 2017. Reference is made to 7,362,493 disability-adjusted life years (DALYs) in 2016; however, the source citation provided was published in 2013. Authors should check and ensure this is right.

The materials and methods lack details in many sections. It would be useful to know the major crops grown by the study population as authors has indicated in the introduction that the few studies conducted Ethiopia on pesticide use only focuses on flower farm workers. Were illness the only exclusion criteria applied in selecting the study population? The procedure for arriving at the calculated sample size would need further explanation. In the qualitative studies, authors indicated the use of different groups of participants. What was the purpose of using these groups in addition to the farmers? How exactly was the study conducted using these groups; did the was the same questionnaire used as in the case of the farmers? Would be useful to see a sample of the questionnaire(s) used. Overall, the materials and methods require reviewing for clarity.

The results will need a general overhaul as in most cases, it is difficult to understand the results presented in-text and how they correspond with what is presented in the tables. Some of the terminologies used by authors will require further explanation. For example, if a participant desires to wash hands after spray; it does not mean the participant washes the hands. There are some important results presented in the tables but not highlighted intext by authors. For instance, storing pesticide in a kitchen, leaving empty containers on farm area etc. is a very serious safety issue but authors have ignored this in their analysis and discussion of the results. It is not clear what the paragraph under Qualitative finding of safety pesticide use practice mean.

There are few citations that are not included in the reference list. Reference list is up to 23, however, there are citations above that e.g., 26, 37 and 40. Authors should ensure all references are cited (in-text) and listed according to the journal requirements.

Overall, there are deficiencies in parts of the manuscripts which impacts on the reliability of the results reported and author have to address these. The manuscript can therefore not be published as it is. Authors will have to apply the suggestions indicated.

6. PLOS authors have the option to publish the peer review history of their article (what does this mean?). If published, this will include your full peer review and any attached files.

Reviewer #1: **Yes: **John Gushit

Reviewer #2: No

---

## [Author Response · Author response to Decision Letter 0]

24 Nov 2022

Authors’ response to reviews

Authors

Fisseha Alebachew (fissehaalebachew9@gmail.com)

Muluken Azage (mulukenag@yahoo.com)

Genet Gedamu (geni_31280@yahoo.com) 

Title: Pesticide use safety practices and associated factors among farmers in Fogera district wetland areas, south Gondar zone, Northwest Ethiopia.

ID: PONE-D-22-18751

Dear Editor, 

All authors have respects for helpful comments and suggestions that the editor and reviewers have made towards the improvements of our manuscript. We believe that we have significantly improved it and we made the necessary corrections after carefully considered and taken all of your comments. Additionally, the revised manuscript also extensively examined to correct grammatical mistakes and spelling inconsistencies. We use “Track Changes Highlights” for all affected revisions and corrections in the “Revised Manuscript”. We have also an unmarked version of our revised manuscript without tracked changes. Finally, a point by point response to the reviewers’ concerns is listed below.

Reviewers’ comments and responses

Reviewer #1: John Gushit

Dear Reviewer #1,

It is a great opportunity for us to receive helpful comments and precious advices from you. With all respect we thank you Sir, for your helpful comments for the improvement of this manuscript. We have carefully considered and taken all of your comments when rewriting the manuscript. Please follow the point by point response.

Comment: The English Language needs to be improved upon

Response: Thank you for the comment it is corrected according to your wise advice. Please see the amendments in all sections of the revised manuscript.

Comment: Needs to get a recent edition of the PLOS ONE Journal to guide in i) presentation of results ii) discussions

Response: Thank you for the comment it is corrected according to your wise advice. Please see the amendments in all sections of the revised manuscript

Comment: Adhere to the PLOS ONE Template for listing references. (Refer to https://journals.plos.org/plosone/s/submission-guidelines)

Response: Thank you for the comment it is corrected according to your wise advice. Please see the citation at introduction and discussion section of the revised manuscript and the reference on the list of reference section of the revised manuscript.

Comment :) Appropriate URL and DOI should be properly inserted

Response: Thank you for the comment it is corrected according to your wise advice. Please see the amendments in the list of reference section of the revised manuscript.

Comment: Appropriate URL and DOI should be properly inserted

Response: Thank you for the comment it is corrected according to your wise advice. Please see the amendments in the list of reference section of the revised manuscript.

Comment: Punctuation

Response: Thank you for the comment it is corrected according to your wise advice. Please see the amendments in all sections of the revised manuscript

Comment: Justification

Response: Thank you for the comment it is corrected according to your wise advice. Please see the amendments in the introduction sections of the revised manuscript

Reviewer ≠2

Comment: The title will need reviewing to exclude 2021

Response: Thank you for the comment it is corrected according to your wise advice. Please see the amendments in the title section of revised manuscript at the first page.

Comment: Manuscript has no line numbering and therefore difficult to provide feedback on specific lines

Response: Thank you for the comment, it is corrected according to your wise advice. Please see the amendments in the whole section of revised manuscript .

Comment: Generally, the manuscript contains lots of grammatical errors, and some sections are unclear and difficult to read. I strongly recommend the authors check the grammar carefully and enhance the clearness of sentences throughout the manuscript.

Response: Thank you for the comment it is corrected according to your wise advice. Please see the amendments in all sections of the revised manuscript.

Comment: Abstract will need reviewing, particularly, on the methods/procedures of the experiment to reflect order of processes presented in the main text. Overly, statistical analysis is what is presented in the methods and not sufficient of how data was collected. Why include 2021 in the objective, which is indicated as aim in the main text. Results is unclear as presented

Response: Thank you for the comment it is corrected according to your wise advice. Please see the amendments in the abstract section of revised manuscript.

Comment: In the introduction, authors have defined pesticides as drug used for controlling and preventing pests, weeds, vectors, rodents, and insects. Is this definition of pesticides as drugs, right? 

Response: Thank you for the comment it is corrected according to your wise advice. Please see the amendments in the introduction section of revised manuscript.

Comment 1: It would also be clearer for authors to indicate the different types of pesticides used for controlling weeds, rodents etc. 

Response: Thank you for the comment it is corrected according to your wise advice. Please see the amendments in the whole introduction section of revised manuscript.

Comment: Most part of the introduction is unclear due to grammatical issues. 

Response: Thank you for the comment it is corrected according to your wise advice. Please see the amendments at paragraph 4 of the introduction section of revised manuscript.

Comment: At the beginning of paragraph 2 of the introduction, authors referred to ‘international bodies taking up the issues and adopting a number of solutions. It is not clear which issues authors were referring to. Can this be clarified? 

Response: Thank you for the comment it is corrected according to your wise advice. Please see the amendments in paragraph 2 of the introduction section at revised manuscript.

Comment 2: Authors also indicated that ‘despite the efforts, global pesticide use has continued to grow steadily to 4.1 million tons per year in 2017. Would be useful for authors to indicate from which year to 2017.

Response: Thank you for the comment it is corrected according to your wise advice. Please see the amendments at paragraph 2 of the introduction section of the revised manuscript

Comment: Reference is made to 7,362,493 disability-adjusted life years (DALYs) in 2016; however, the source citation provided was published in 2013. Authors should check and ensure this is right.

Response: Thank you for the comment it is corrected according to your wise advice. Please see the amendments in the list of reference section of revised manuscript.

Comment: The materials and methods lack details in many sections. It would be useful to know the major crops grown by the study population as authors has indicated in the introduction that the few studies conducted Ethiopia on pesticide use only focuses on flower farm workers.

Response: Thank you for the comment it is corrected according to your wise advice. Please see the amendments in the study setting and period section of the revised manuscript.

Comment: Were illness the only exclusion criteria applied in selecting the study population?

Response: Thank you for the comment. Yes, we used this as the exclusion criteria. A farmer with any type of illness that prevent him from provision of information regarding to the research during the time of data collection were excluded from the study. 

Comment: The procedure for arriving at the calculated sample size would need further explanation. 

Response: Thank you for the comment it is corrected according to your wise advice. Please see the amendments in the Sample size determination section of the revised manuscript

Comment: In the qualitative studies, authors indicated the use of different groups of participants. What was the purpose of using these groups in addition to the farmers? 

Response: Thank you for the comment. The main purpose of using key informant and in-depth interview participants were to explore and dig out more barriers of pesticide use safety practices. Pesticide use safety practice is not only the role of farmers. Many stalk holders have a role on pesticide use safety practice. To increase the pesticide safety practice by the farmers the role of many stakeholders is very crucial by creating awareness, providing personal protective equipment, applying and enforcing laws and others. Therefore, it is very difficult to address all of these issues by only quantitative data from the farmers. That is why we use qualitative data to explore more challenges from selected responsible stakeholders.

Comment: How exactly was the study conducted using these groups; did there was the same questionnaire used as in the case of the farmers? Would be useful to see a sample of the questionnaire(s) used. 

Response: Thank you for the comment. We used different questionnaire from the farmer. We will submit the questionnaire that we used with the revised manuscript by entitling S2 Survey tool as per your direction. 

Comment: The results will need a general overhaul as in most cases; it is difficult to understand the results presented in-text and how they correspond with what is presented in the tables. Some of the terminologies used by authors will require further explanation. For example, if a participant desires to wash hands after spray; it does not mean the participant washes the hands. There are some important results presented in the tables but not highlighted in text by authors. For instance, storing pesticide in a kitchen, leaving empty containers on farm area etc. is a very serious safety issue but authors have ignored this in their analysis and discussion of the results. 

Response: Thank you for the comment it is corrected according to your wise advice. Please see the amendments in the whole result section of revised manuscript.

Comment: It is not clear what the paragraph under Qualitative finding of safety pesticide use practice mean.

Response: Thank you for the comment. We tried to rewrite it this section to make clearer for readers. The whole paragraph in this section states the possible challenges for pesticide safety practice obtained from the key informant and in-depth interview that supports the quantitative data obtained from the farmer. 

Comment: There are few citations that are not included in the reference list. Reference list is up to 23; however, there are citations above that e.g., 26, 37 and 40. Authors should ensure all references are cited (in-text) and listed according to the journal requirements

Response: Thank you for the comment it is corrected according to your wise advice. Please see the citation at introduction and discussion section of the revised manuscript and the reference on the list of reference section of the revised manuscript.

---

## [Editor Report · Decision Letter 1]

22 Dec 2022

Pesticide use safety practices and associated factors among farmers in fogera district wetland areas, south Gondar zone, Northwest Ethiopia 2021

PONE-D-22-18751R1

Dear Dr. Kassie,

We’re pleased to inform you that your manuscript has been judged scientifically suitable for publication and will be formally accepted for publication once it meets all outstanding technical requirements.

Kind regards,

Haruna Musa Moda

Academic Editor

PLOS ONE

Additional Editor Comments (optional):

Dear Authors

Many thanks for taking time out to effect all corrections recommended.

I am happy to recommend the manuscript be accepted in its present form.

Best wishes

Haruna
---

## [Editor Report · Acceptance letter]

2 Jan 2023

PONE-D-22-18751R1 

Pesticide use safety practices and associated factors among farmers in Fogera district wetland areas, south Gondar zone, Northwest Ethiopia 

Dear Dr. Kassie:

I'm pleased to inform you that your manuscript has been deemed suitable for publication in PLOS ONE. Congratulations! Your manuscript is now with our production department. 

Kind regards, 

on behalf of

Dr. Haruna Musa Moda 

Academic Editor

PLOS ONE